# Bacterial Communities Related to Aroma Formation during Spontaneous Fermentation of ‘Cabernet Sauvignon’ Wine in Ningxia, China

**DOI:** 10.3390/foods11182775

**Published:** 2022-09-09

**Authors:** Zhong Zhang, Qingchen Zhang, Hui Yang, Lijun Sun, Hongchuan Xia, Wenjing Sun, Zheng Wang, Junxiang Zhang

**Affiliations:** 1School of Life Sciences, Ningxia University, Yinchuan 750021, China; 2College of Pharmacy, University of Florida, Gainesville, FL 32610, USA; 3School of Agriculture, Ningxia University, Yinchuan 750021, China; 4School of Food & Wine, Ningxia University, Yinchuan 750021, China; 5Engineering Research Center of Grape and Wine, Ministry of Education, Yinchuan 750021, China

**Keywords:** Ningxia, Cabernet Sauvignon, wine, spontaneous fermentation, bacteria, flavor

## Abstract

Bacteria are an important part of wine ‘microbial terroir’ and contribute to the formation of wine flavor. Based on high-throughput sequencing and non-targeted metabonomic technology, this study first explored the bacterial composition and its effect on the aroma formation of spontaneously fermented ‘Cabernet Sauvignon’ (CS) wine in the Eastern Foot of Helan Mountain (EFHM), Ningxia. The results showed that there were significant differences in bacterial communities during fermentation of CS grapes harvested from different sub-regions of EFHM, with the earlier-established vineyard obtaining more species. The level of bacterial diversity initially decreased and then increased as the fermentation proceeded. Malolactic fermentation (MLF) was spontaneously initiated during alcohol fermentation (AF). *Pantoea*, *Lactobacillus*, *Rhodococcus*, *Fructobacillus*, and *Komagataeibacter* were the core bacterial genera in the fermentation mixture. *Lactobacillus* contributed to the synthesis of methyl and isobutyl esters and the formation of red and black fruity fragrances of wine. *Fructobacillus* was closely related to the synthesis of aromatic alcohols and the generation of floral flavors.

## 1. Introduction

Wine exhibiting distinctive geographical characteristics is an indispensable aspect of consumer preference and economic appreciation [1]. It is accepted that the climate (macroclimate, mesoclimate, and microclimate), soil (geology and pedology), cultivar (grapevine and rootstock), as well as anthropogenic factors (history, culture, agronomic management, and brewing craft), collectively known as ‘terroir’, work together to determine the organoleptic distinctiveness of wine from a particular region [2]. Moreover, these elements may condition what has been defined as the ‘microbial terroir’, since grape microflora displays a nonrandom distribution pattern across different viticultural zones and are associated with fruit health and wine phenotypes [3,4,5]. Specifically, the “microbial terroir” is a combination of microbial taxa of a given region, consisting of autochthonous fungi and bacteria that can confer specific regional character in wines [1,4].

AF is a complex biochemical process driven by microorganisms. Yeast (e.g., *Saccharomyces*, *Torulaspora*, *Schizosaccharomyces*, *Pichia*, *Lachancea*, *Metschnikowia*, and *Hanseniaspora*) plays a leading role in the AF process by converting sugars into ethanol and carbon dioxide and generating a slew of volatile organic compounds (VOCs), such as higher alcohols, esters, fatty acids, terpenes, and varietal thiols, which are essential to wine aroma [6]. Some filamentous fungi, primarily the genera *Alternaria*, *Aspergillus*, *Botryotinia*, *Cladosporium*, *Davidiella*, and *Penicillium*, also exist in grape juice, producing monoterpenoids from geranyl pyrophosphate during AF [7,8,9]. However, many of them decrease rapidly with the progress of fermentation, resulting in a limited contribution to wine flavor [7,9]. Compared with fungi, the level of bacterial diversity found in vineyards and wineries is higher [9]. Although bacteria cannot dominate AF, some lactic acid bacteria (LAB, e.g., *Oenococcus*, *Lactobacillus*, *Leuconostoc*, and *Pediococcus*) and acetic acid bacteria (AAB, e.g., *Acetobacter* and *Gluconobacter*) can survive under several stresses including sugar-induced high osmolarity, elevated ethanol concentration, as well as low pH, and have a certain impact on the sensory quality of the wine [9,10]. However, there are many other bacterial genera belonging to at least 25 phyla, four of which (*Proteobacteria*, *Firmicutes*, *Actinobacteria*, and *Bacteroidetes*) were predominant, who have been detected in vineyard and winery ecosystems but whose evolutionary patterns during AF and effect on wine aroma are not clear [4,11,12,13,14,15,16].

Yeasts can be easily categorized through special mediums, such as Wallerstein Laboratory (WL) nutrient agar or lysine agar, based on colony morphology [17,18]; however, this method is impractical for bacteria. Although bacterial strains can be identified by DNA extraction, polymerase chain reaction (PCR), and Sanger sequencing platform, it still relies on culture-dependent technology, moreover, it is time-consuming and laborious [19]. Modern molecular biotechnology such as denatured gradient gel electrophoresis (DGGE), fluorescence in situ hybridization (FISH), ribosomal intergenic spacer analysis (RISA), and microsatellite (MSI), can distinguish some unculturable bacteria in wine. However, these methods often fail to provide detailed taxonomic annotation and have a poor response to low-abundance species [14,20,21]. High throughput sequencing (HTS), also known as the next-generation sequencing technology, has the advantages of simplicity, high throughput and high sequencing speed, and a powerful detection mechanism for minor and unculturable bacterial taxa. Furthermore, the data obtained from HTS can be annotated for operational taxonomic units (OTUs) and functions. Therefore, it has been widely used to investigate the microbial ecology of various wine regions in recent years [22,23,24,25].

The EFHM in Ningxia is one of China’s Wine Geographical Indication Product Protection Areas, as it houses a plethora of microbial resources. In recent years, some studies have analyzed the fungal diversity in the wine fermentation broth, berry surface, and rhizosphere soil [26,27,28,29,30] and the bacterial community in fruit surface and vineyard soil [25,29,30] in EFHM; however, research on the dominant bacteria in the fermentation process is sparse. Consequently, the objectives of this study were (1) to investigate bacterial community succession during spontaneous fermentation of a widespread cultivar (*Vitis vinifera* L. cv. Cabernet Sauvignon) in EFHM through the HTS technique; (2) to demonstrate the relationship between core bacteria and wine aroma that was measured by a non-targeted gas chromatography-mass spectrometry (GC-MS) method and quantitative descriptive analysis (QDA).

## 2. Materials and Methods

### 2.1. Grapes

Cabernet Sauvignon grapes were harvested on 26 September 2020 from vineyards located in three main sub-regions of EFHM in Ningxia (Appendix A), labeled Yinchuan (YC, 106°05′ E, 38°56′ N, 4-year vineyard), Yuquanying (YQY, 106°08′ E, 38°26′ N, 15-year vineyard), and Qingtongxia (QTX, 105°88′ E, 38°08′ N, 7-year vineyard), with the sugar content of 239.1/210.5/226.6 g/L, titratable acidity of 4.3/5.6/4.4 g/L (GB/T 15038-2006, 2006) [31], yeast assimilable nitrogen (YAN) of 238.0/249.2/260.4 mg N/L [32], and pH 3.50/3.30/3.55, respectively.

### 2.2. Spontaneous Fermentation and Sample Collection

Grape bunches were selected, destemmed, and crushed under aseptic conditions, without the addition of sulfur dioxide and pectinases. In triplicate, grape musts were spontaneously fermented at 22 to 25 °C in a sterilized 10 L glass bottle in an ultraviolet-irradiated room, where wine elaboration had never been carried out before, thus the spontaneous fermentation would not be cross-contaminated by commercial brewing microorganisms. A water-trap apparatus containing concentrated H_2_SO_4_ was attached to the top of each bottle to trap the water evaporating from the flask during fermentation. Cap management was conducted twice a day using sterile gloves to homogenize the fermentation broth and prevent the growth of spoilage microorganisms on the surface of the cap.

The fermentation course was monitored by measuring the weight loss of each bottle every 12 h (The amount of CO_2_ produced was indirectly measured as the decrease in the weight of the whole flask), as described by Wang et al. (2019) [22]. The predicted alcohol content was calculated according to the reaction equation of AF (Equation (1)). Samples were collected at 0 d (A stage), 1 d (B stage), 2 d (C stage), 4 d (D stage), 8 d (E stage), and 12 d (F stage), immediately quenched by liquid nitrogen and stored at –80 °C for HTS and VOCs analysis.
(1)$$C_{6}H_{12}O_{6}+2\mathrm{Pi}+2\mathrm{ADP}+2H^{+}\to 2{\mathrm{CH}}_{3}{\mathrm{CH}}_{2}\mathrm{OH}+2{\mathrm{CO}}_{2}+2\mathrm{ATP}+2H_{2}O$$
where Pi is a phosphate ion; ADP is adenosine diphosphate; ATP is adenosine triphosphate.

### 2.3. DNA Extraction and PCR Reaction

Genomic DNA was extracted using the E.Z.N.A. soil DNA Kit (Omega Bio-Tek, Norcross, GA, USA) according to the manufacturer’s protocols. The quality of DNA was assessed by 1% agarose gel electrophoresis and a NanoDrop spectrophotometer (Thermo Fisher Scientific, Wilmington, DE, USA). The 16S rRNA V3–V4 regions of bacteria were amplified using PCR primers 338F (5′-ACTCCTACGGGAGGCAGCAG-3′) and 806R (5′-GGACTACHVGGGTWTCTAAT-3′) [33]. The reaction mixture contained 4 μL 5 × Fast Pfu buffer, 2 μL dNTPs (2.5 mM), 0.8 μL forward primer (5 µM), 0.8 μL reverse primer (5 µM), 0.4 μL Fast Pfu polymerase, 0.2 μL BSA, 10 ng template DNA, and finally ddH_2_O up to 20 μL. The amplification procedures were 95 °C for 3 min, 27 cycles of 95 °C for 30 s, 55 °C for 30 s, 72 °C for 45 s, and a final extension at 72 °C for 10 min.

### 2.4. HTS Analysis

The PCR products were purified using the AxyPrep DNA Gel Extraction Kit (Axygen Biosciences, Union City, CA, USA) and quantified using a Quantus™ Fluorometer (Promega, Fitchburg, WI, USA). The purified amplicons were pooled in equimolar and paired-end sequenced on an Illumina MiSeq PE300 platform (Illumina, San Diego, CA, USA) in Majorbio Bio-pharm Technology Co., Ltd. (Shanghai, China). The raw data were submitted to the NCBI Sequence Read Archive (SRA) repository with the accession number SRP361551.

### 2.5. VOCs Analysis

VOCs were analyzed using the headspace solid-phase microextraction (HS-SPME) coupled with GC-MS according to the general protocol of our laboratory [34]. The identification of VOCs was accomplished by matching the obtained mass spectra with the NIST 17 standard library and by comparing the retention indices (RIs), which were calculated based on the retention times of C8–C40 alkanes (Sigma-Aldrich, Shanghai, China), to those reported in the NIST Chemistry WebBook (https://webbook.nist.gov/ (accessed on 26 July 2022)) (Appendix A). Considering the bacteria were analyzed by relative abundance, the VOCs were quantified by the percentage of peak area (Equation (2)) [35]:(2)$${\mathrm{RC}}_{i}=\frac{A_{i}/A_{\mathrm{IS}}}{\sum_{i=1}^{n}A_{i}/A_{\mathrm{IS}}}\times 100\%$$
where RC_i_ is the relative concentration of each VOC; A_i_ is the peak area of each VOC; A_IS_ is the peak area of the internal standard.

### 2.6. Sensory Analysis of Aroma Profile

Aroma characteristics of spontaneously fermented wines were evaluated by a panel of 19 judges (Ten females and nine males, 20–30 years of age, with at least three years of wine-tasting experience). The group members were trained with the ‘Le Nez du Vin’ aroma kit (Ease Scent, Beijing, China) over four weeks before taking part in the formal sniffing. The training was carried out three times each week for 45–60 min. At the end of the fourth week, a copy of wine samples was served to the panelists for assessment and discussion. In total, 12 aroma descriptors, including red fruits, black fruits, tropical fruits, drupes, nuts, jams, dried fruits, flowers, green grass, mesothecium, spices, and cream, were chosen for QDA.

In the formal sensory analysis, wine samples were served in random order in covered tasting glasses (ISO 3591–1997), and in a standard tasting room (ISO 8589–1998) at room temperature. Each of the panelists was asked to score the intensity of each descriptor on a 0–10 scale two times. The accuracy and repeatability of each person were assessed through the Panel Check software (Version 1.4.2, Nofima Mat, Tromsø, Troms, Norway), and finally, 14 panelists passed the checking (Appendix A). Modified frequency (MF) was calculated for aroma evaluation (Equation (3)) [36]:(3)$$\mathrm{MF}=\sqrt{F\times I}$$
where F is the perceived frequency of each descriptor, %; I is the average intensity of each descriptor, %.

### 2.7. Bioinformatics Analysis and Statistical Analysis

The raw 16S rRNA gene sequencing reads were demultiplexed, quality filtered by Fastp (version 0.20.0, Haplox, Shenzhen, Guangdong, China), and merged by FLASH (version 1.2.7, Center for Computational Biology, Baltimore, MD, USA). OTUs with a 97% similarity cutoff were clustered using uParse (version 7.1, Independent Investigator, Tiburon, CA, USA), and chimeric sequences were identified and removed. The taxonomy of each OTU representative sequence was analyzed by RDP Classifier (version 2.2, Center for Microbial Ecology, East Lansing, MI, USA) against the Silva database using a confidence threshold of 0.7.

All statistical analyses were performed by R packages (version 4.2.0, R Foundation for Statistical Computing, Vienna, Austria) and OriginPro 2022 software (OriginLab Corporation, Northampton, MA, USA). A non-metric multidimensional scaling (NMDS) plot was created to illustrate the separation of different samples. Random forest (RF) analysis was used to determine the core bacterial genus. Wilcoxon rank sum test and Kruskal–Wallis H test were applied to determine the variance of alpha diversity indices among different sub-regions and different fermentation stages. Variance inflation factor (VIF) analysis was employed to filter VOCs with multicollinearity. A mantel test was performed to investigate the integral relationship between filtered VOCs and core bacteria. Redundancy analysis (RDA) or correlation heatmap analysis was utilized to visualize the one-to-one relationship between each VOC and core bacteria, and the tolerance of core bacteria to ethanol.

## 3. Results

### 3.1. Bacterial Composition of Different Sub-Regions

The fermentation rates of Cabernet Sauvignon grapes from different sub-regions are shown in Figure 1a. The three fermentations had similar patterns, where the vigorous period of AF began at 2 d (C stage), terminated at 8 d (E stage), and the final CO_2_ releasement at 12 d (F stage) was 118.6 g/L for YC, 103.4 g/L for YQY, and 110.3 g/L for QTX, respectively. The predicted alcohol concentration calculated based on CO_2_ production was 15.7 (*v/v*, %) for YC, 13.7 (*v/v*, %) for YQY, and 14.6 (*v/v*, %) for QTX. The actual alcohol content measured according to the National Standard of China (GB/T 15038-2006, 2006) was 15.0 (*v/v*, %) for YC, 13.3 (*v/v*, %) for YQY, and 13.9 (*v/v*, %) for QTX (Appendix A). Gaps between the predicted and real values were within 1.0 (*v/v*, %).

A total of 2,344,803 high-quality bacterial V3–V4 Illumina sequences were obtained from 54 samples, and the average length was 417 bp. After further quality control of the raw reads and removal of chloroplast and mitochondrial sequences, the clean tags were clustered into 1,689 OTUs, belonging to 750 genera, 386 families, 238 orders, 103 classes, and 35 phyla. The rarefaction curve tended to be flat, and the coverage of bacteria exceeded 95% (Figure 1b), indicating that the depth of sequencing was sufficient to represent the microbial diversity in each sub-region [37]. Although the curves of the observed OTUs were not parallel to the *x*-axis (Figure 1c) and those of shared OTUs flattened (Figure 1d); thus, it was hopeful to determine core bacteria in this region.

Bacterial community dissimilarities among different samples are shown by NMDS plots based on taxonomic (Bray–Curtis) and the phylogenetic (weighted UniFrac) distance (Figure 2) [38]. Taking the 18 samples in each sub-region, as a whole, the variation among different sub-regions was significant (stress < 0.1, *p* < 0.001). In addition, for different fermentation stages, stage A/B in YC and YQY, and stage A in QTX, can be separated from other stages, indicating that the composition of the bacterial community may change once the fermentation started.

Illumina MiSeq sequencing platform can obtain more accurate annotation information at the genus level than at the species level. Therefore, the alpha diversity based on genera in three sub-regions was measured using non-phylogenetic (including Chao 1, Shannon, and Shannoneven index) and phylogenetic (Faith’s phylogenetic diversity, PD) indices [38]. According to Figure 3a–d, YQY had the highest levels of the four alpha diversity indices above, YC had the lowest values of Chao 1 and PD indices, and QTX had the lowest Shannon and Shannoneven indices. It could be further seen from Figure 3e that YQY not only had the most total number of bacterial genera (644) but the largest number of exclusive genera (294). In addition, there were 181 bacterial genera shared by the three sub-regions.

Among the 181 bacterial genera, 10 had a relative abundance of more than 1%, accounting for 89.21% of the total amount, while the remaining 171 genera only occupied 10.79%. Among the top 10 genera (Figure 3f), *Pantoea* had the highest relative abundance of 40.89%, followed by *Lactobacillus* with 21.88%.

A RF supervised-learning model was used to reveal the core bacteria that could explain the strongest variation among samples from different sub-regions (Figure 4a) [3,33]. Seven bacteria genera were selected as biomarkers (Figure 4a), of which *Pantoea*, *Lactobacillus*, *Fructobacillus*, *Komagataeibacter*, and *Rhodococcus* were shared by the three sub-regions, with a relative abundance higher than 1% (Figure 3f). Therefore, they were preliminarily determined as the core bacteria in the spontaneous fermentation process of Cabernet Sauvignon wine in Ningxia. Figure 4b further showed their distribution in different sub-regions. 72.61% of *Pantoea* was from QTX, 99.79% of *Komagataeibacter* was from YQY, while *Lactobacillus*, *Rhodococcus*, and *Fructobacillus* were mainly from YC.

### 3.2. Bacterial Composition of Different Fermentation Stages

According to the previous analysis (Figure 2), the bacterial community composition may be different not only among three sub-regions but also among different stages of fermentation. Furthermore, Table 1 suggested that during fermentation the alpha diversity (Chao 1, Shannon, Shannoneven, and PD indices) of bacteria initially decreased and then increased. The overall variation of different fermentation stages was significant (*p* < 0.05), and the highest values of the four indices were mostly at stage A (except for the Chao 1 and PD indices of YQY). The lowest values of Chao 1 and Shannon indices were both at the D stages of YC and YQY, and the C stage of QTX. The lowest values of Shannoneven and the PD index ranged from C to E stages. Therefore, the grape juice/must (stage A) always had the most complex bacterial composition, most of them may be inhibited during the exuberant phase of AF (stages C–E), but the inhibitory effect would be relieved at the end of AF (stage F).

Figure 5a showed the successions of bacteria with relative abundance greater than 0.1% during the AF process, and Figure 5b showed the changes of the five core bacteria. *Lactobacillus* is one of the main LAB involved in the MLF of wine. In YC, *Lactobacillus* began to increase when the fermentation started, concentrated at C–F stages, with relative abundance greater than 70%. In addition, a certain amount of *Lactobacillus* was also detected in the vigorous and late fermentation stages of YQY and QTX. The other two important genera of YC were *Rhodococcus* and *Fructobacillus*. The former was concentrated in A, B, E, and F stages, indicating that it may be subject to some competitive inhibitions during the vigorous period of AF. The latter was mainly distributed in the B–F stages in YC and was also enriched in C–F stages during YQY and QTX fermentations.

As for the YQY sub-region, the core AAB taxa *Komagataeibacter* was mainly distributed in the A–C stages, and then decreased rapidly. Moreover, *Komagataeibacter* only existed in the A stage of YC and the E stage of QTX. Another acetic bacterium, *Gluconobacter*, was also found in the early stages of YQY fermentation broth. Therefore, as the fermentation proceeded, their negative impacts on wine aroma quality may be limited. At the F stage of YQY, a large number of *Oenococcus* (relative abundance of 59.19%) were detected. *Oenococcus* is another kind of LAB leading the MLF process of wine, with *O. oeni* as the type species.

For the QTX sub-region, the dominant bacterial genera in stage A were *Massilia*, *Pseudomonas*, and *Sphingomonas*. However, once fermentation was initiated, *Pantoea* rapidly dominated the population, with a relative abundance higher than 88.33% from stage B to F. In addition, *Pantoea* was also the most dominant genus in stage D and stage E in YQY, with relative abundance reaching 75.96% and 41.22%, respectively.

The FAPROTAX (functional annotation of prokaryotic taxa) database was used to annotate the bacterial community composition at different fermentation stages (Figure 5c). These results have high confidence as the FAPROTAX database is constructed based on current literature of available culturable strains which maps prokaryotes to established metabolic pathways [39]. It can be seen from Figure 5c that the bacteria during AF of Cabernet Sauvignon were mainly ‘chemoheterotrophic’ and therefore very dependent on the nutrients in the fermentation matrix. The differences in the carbon and organic nitrogen sources of grapes may be the reason for the dissimilarities in bacterial communities of the three sub-regions. ‘Fermentation’ was also one of the key functions of bacteria, and the abovementioned assumption that both *Lactobacillus* and *Oenococcus* species undergo MLF during AF could be verified by comparing Figure 5c with Figure 5a. The functions of ‘hydrocarbon degradation’ and ‘aromatic compound degradation’ may be associated with the production of some VOCs in wine. In general, the main types of microorganisms that consume nitrogen sources from wine juice are fungi, especially yeasts [32]. That said, the bacteria in the fermentation matrix in this study were also found to have certain nitrogen metabolism functions, including ‘nitrate reduction’, ‘nitrogen respiration’, ‘nitrate respiration’, and ‘nitrite respiration’ (Figure 5c).

### 3.3. Correlation Analysis between Core Bacteria and Wine Aroma/Ethanol

To further investigate the bacterial function in the wine aroma formation, 72 VOCs were detected by HS-SPME-GC-MS and classified into 17 categories based on the similarity of chemical structure and aroma characteristics (Appendix A). Their evolutionary patterns during the AF process are shown in Figure 6.

In situations where variables in the regression model are highly correlated and begin to exhibit multicollinearity, coefficient estimates for individual predictors may become inaccurate [40]. This diagnostic is the reciprocal of the more common VIF, where a value of less than 10 indicates multicollinearity is not an issue [41]. Finally, 13 categories of VOCs were selected for further analysis (Table 2).

The Mantel test was performed to assess the significance of Spearman’s correlation between the two matrices: the relative concentrations of VOCs and the relative abundance of core bacteria. It is shown in Table 3 that both the coefficient and significance level improved after VIF filtration, regardless of whether the computation strategy used was Euclidean distance or Manhattan distance. Therefore, the VIF-filtered 13 categories of VOCs were better suited for later analysis.

The one-to-one relationships between each of the 13 VOC categories and the core bacteria are shown in Figure 7. RDA is a constrained ordination method of summarizing linear relationships in a set of dependent variables influenced by a set of independent variables, using a blend of multiple linear regression and principal components analysis [42]. For the YC sub-region, the first two principal components of the RDA biplot cumulatively explained 86.41% of the variables. *Pantoea* and *Rhodococcus* had similar evolutionary patterns during fermentation, and *Lactobacillus* and *Fructobacillus* had strong concordance (Figure 7a). According to the correlation heatmap (Figure 7b), *Pantoea* was significantly and positively correlated with straight-chain fatty alcohols, aromatic aldehydes, and terpenes; *Rhodococcus* was significantly and positively correlated with aromatic aldehydes and terpenes; *Lactobacillus* was significantly and positively correlated with acetic esters, methyl esters, isobutyl esters, branched chain fatty alcohols, aromatic alcohols, fatty acids, and sulfides; *Fructobacillus* was significantly and positively correlated with acetic esters, branched-chain fatty alcohols, aromatic alcohols, fatty acids, and sulfides. Although *Komagataeibacter* was well correlated with aromatic alkenes (Figure 7a), the correlativity was not significant (Figure 7b).

The cumulative explanation of the first two principal components of the RDA analysis for the YQY sub-region was 94.65% (Figure 7c). *Pantoea*, *Lactobacillus*, and *Fructobacillus* were significantly and positively correlated with methyl esters, isobutyl esters, isoamyl esters, and aromatic alcohols, and *Pantoea* also contributed to the production of acetic acids and branched-chain fatty alcohols during fermentation (Figure 7d). In addition, the core bacterium *Komagataeibacter* in YQY was significantly and positively correlated with straight-chain fatty alcohols, fatty aldehydes, and aromatic aldehydes (Figure 7d).

In QTX, the first and second principal components of the RDA analysis accumulated 95.48% of the explanatory variables (Figure 7e). The core genus *Pantoea* was significantly and positively correlated with acetic acids, straight-chain fatty alcohols, and fatty acids. Both *Lactobacillus* and *Fructobacillus* were significantly and positively correlated with methyl esters, isobutyl esters, isoamyl esters, aromatic alcohols, and sulfides. Moreover, *Fructobacillus* also played an important role in the generation of acetic acids and branched-chain fatty alcohols (Figure 7f). However, *Rhodococcus* and *Komagataeibacter* contributed less to the synthesis of VOCs (Figure 7e,f).

Overall, in all the sub-regions, the succession of *Lactobacillus* during spontaneous fermentation was significantly and positively correlated with the production of methyl esters, isobutyl esters, and aromatic alcohols, whereas *Fructobacillus* was significantly and positively correlated with aromatic alcohols.

To test the above results of RDA analysis and correlation heatmap, the aroma profiles of spontaneously fermented Cabernet Sauvignon wines were evaluated by QDA. It can be seen from Figure 8a that the olfactory intensities of red fruits and flowers were more prominent in the YC wine. The typical flavor profiles of the YQY wine were tropical fruits, jams, and green grass. The QTX wine aroma was described as nuts and dried fruits.

The correlation analysis between aroma characteristics and core bacteria (Figure 8b) showed that *Lactobacillus* and *Rhodococcus* significantly and positively correlated with red and black fruits. Interestingly, *Lactobacillus* contributed to the production of methyl esters and isobutyl esters (Figure 7), which could provide the wine with fruity odors [16]. Moreover, the bacterial genus *Fructobacillus*, which may generate aromatic alcohols during AF (Figure 7), was significantly and positively correlated with the descriptor of flowers [43].

During wine fermentation, ethanol continues to accumulate and may have a considerable inhibitory effect on some bacteria [44]. As seen in Figure 9a, the changes in the relative abundances of both *Pantoea* and *Rhodococcus* during YC fermentation were significantly and negatively correlated with the increase in ethanol, suggesting that higher alcohol content may be one of the reasons for the decrease of these bacteria. However, *Pantoea* was not sensitive to ethanol stress in the YQY fermentation broth (Figure 9b), nor was *Rhodococcus* in QTX (Figure 9c). Although *Komagataeibacter* was sensitive to ethanol in the YC and YQY fermentations, a similar response was not observed during QTX fermentation (Figure 9b,c). Therefore, the responses of the above three core bacterial genera to ethanol stress needed further exploration.

Notably, the evolutionary patterns of *Lactobacillus* and *Fructobacillus* were significantly and positively correlated with the accumulation of ethanol in all fermentations of grapes from three sub-regions (Figure 9a–c), indicating that they were not suppressed by ethanol and remained active at the late stages of AF.

## 4. Discussion

Given that the key bacteria and their functions in the fermentation process of wine from the Eastern Foot of Helan Mountain in Ningxia were not well understood, we harvested raw grapes of the main cultivar ‘Cabernet Sauvignon’ from different sub-regions for spontaneous fermentation. Despite not being inoculated with commercial yeasts, the indigenous microorganisms were able to complete alcohol fermentation (wine with residual sugar content below 4 g/L) (Appendix A). However, the fermentation rates of the various sub-regions differed, most likely due to the differences in the YAN of grapes [22,45], showing that the grape with higher YAN content (in QTX) was able to reach the exuberant period earlier (Figure 1a). This also had an impact on the bacterial composition during fermentation. On the one hand, the NMDS model distinguished samples before the vigorous stage (stages A and B) from other stages (Figure 2), and previous studies also showed that the bacterial community would change once the fermentation was initiated [33]. Moreover, the bacterial diversity indices dropped to the lowest level during the logarithmic period of yeasts (Table 1), probably because most bacteria would be confronted with nutritional competition from the dominant fungi (yeasts) during this time [46]. The lack of nitrogen sources would directly lead to decreased activity of bacteria since the bacterial functions annotated by the FAPROTAX database in this study were associated with multiple nitrogen metabolism (Figure 5c).

In a particular wine region, the ‘microbial terroir’ is formed gradually over time, influenced by a variety of factors such as local climate, soil, flora, and cultivation practices, thus older vineyards may have a more complex microecology [3,38,46]. In this study, the grapes obtained from a 15-year vineyard in the YQY sub-region showed higher bacterial diversity during fermentation than grapes from the other two younger vineyards. However, the great exclusive genera found (294) in YQY only had a relative abundance of 1.84% in this sub-region, suggesting a limited contribution to the “microbial terroir” of wine (Appendix A).

The machine learning method of random forest analysis revealed the important role of five bacterial genera, *Pantoea*, *Lactobacillus*, *Rhodococcus*, *Fructobacillus*, and *Komagataeibacter*, in the spontaneous fermentation of ‘Cabernet Sauvignon’ wines (Figure 4a). Regarding the most abundant bacteria, microorganisms from the genus *Pantoea* were detected in all the sub-regions, enriched in QTX samples, and rapidly dominated the population after fermentation initiation (Figure 5a). Studies have shown that *Pantoea* is also a key feature in other wine regions, such as Penglai, Yangling, Shanshan, Yanqi, Heshuo, Huoerguosi, Fukang, and Manasi in China [25,27]; Minho, Douro, Dão, Bairrada, Estremadura, and Alentejo in Portuguese [8]; Modra in Slovakia [47]; Central Pennsylvania in the United States [48]; moreover, the Ullum Valley of San Juan Province in Argentina [49]. The predominant presence of this group is independent of the identification methods. For example, we used PCR primers 338F/806R to amplify the 16S rRNA V3–V4 regions of bacteria and an Illumina MiSeq PE300 platform for sequencing. Similar results were obtained by amplifying the 16S rRNA V4 region using primers 515F/806R and sequencing on an Illumina HiSeq 2500 platform [27], or through amplifying the V6 hypervariable region of 16S rRNA and sequencing on a 454 Titanium platform [8]. In this study, *Pantoea* was found to have a significant positive correlation with a few categories of VOCs during fermentation, such as higher alcohols (straight-chain fatty alcohols in YC and QTX, and branched-chain fatty alcohols in YQY), esters (acetic esters, methyl esters, isobutyl esters and isoamyl esters in YQY, and acetic esters in QTX), and fatty acids in QTX. This is in line with the previous reports [30,48,50], but there was no consistent pattern among different sub-regions (Figure 7).

Recently, Zheng et al. (2020) [51] proposed the reclassification of the genus *Lactobacillus* into 25 new genera based on whole genome sequences. However, the obtained sequences in this work were analyzed against the Silva 138.0 database, thus we still use the old taxonomic name of *Lactobacillus*. This genus was mainly found in YC samples and dominated the population from the start to the end of alcoholic fermentation. Another lactic acid bacterium, *Oenococcus*, was detected in the late stage of YQY fermentation (Figure 5a), which indicates that MLF has been spontaneously initiated during AF. This was also proved in Appendix A, where some malic acids were transferred into lactic acids. However, the rate of MLF was different in each wine, and only a small amount of malic acid was consumed in QTX wine samples, due to a lower relative abundance of LAB. We also noticed that the production of lactic acid in YC and QTX was greater than the decrease of malic acid, which came with the opposite results in YQY. The reason may be that some lactic acid could be generated through the Embden–Meyerhof–Parnas (EMP) pathway [52], and part of the malic acid could be consumed by other wine-associated bacteria [16,53]. Traditionally, MLF is carried out after the accomplishment of AF by inoculating commercial lactic acid bacteria, while in recent years many wineries have adopted MLF with parallel inoculation fermentation with AF. In most cases of spontaneous fermentation, wild LAB-dominated MLF was able to proceed simultaneously with wild yeast-dominated AF [54]. *Lactobacillus* and *Oenococcus* can convert *L*-malic acid into *L*-lactic acid, improve the microbial stability and taste of wine, and produce glycosidase, esterase, and protease, which indirectly increase the content of some VOCs, enhancing the complexity of wine aroma [53]. Studies have shown that some strains of *Lactobacillus* were associated with the production of aromatic alcohols, branched-chain fatty alcohols, ethyl esters, acetic esters, medium-chain fatty acids, C13-norisoprenoids, terpenes, and volatile phenols [53,55,56]. In this study, we found that *Lactobacillus* may be involved in the synthesis of methyl esters and isobutyl esters in wine (Figure 7).

*Rhodococcus* and *Fructobacillus* were also enriched in YC samples, but no studies have yet shown that *Rhodococcus* affects wine aroma. *Fructobacillus* has been found in some wineries in Spain [21,57] and Napa Valley in the United States [11], which is a group of bacteria capable of breaking down fructose and producing lactic acid, but the metabolic pathway and the impact on wine quality are not known [58]. In this study, *Fructobacillus* was found, for the first time, to be closely associated with the synthesis of aromatic alcohols during the spontaneous fermentation of ‘Cabernet Sauvignon’ wines and to contribute to the floral aroma of the wine (Figure 7). We noticed that *Lactobacillus* and *Fructobacillus* shared some properties of production of aromatic alcohols, resistance to ethanol, and generation of lactic acid. Interestingly, *Fructobacillus* spp. belong to the family *Leuconostocaceae*, and were originally classified as *Leuconostoc* spp., but were later grouped into a novel genus, based on their phylogenetic position, morphologies, and specific biochemical characteristics [58].

*Komagataeibacter* was established as a new acetic bacteria genus deriving from the genus *Gluconacetobacter*, which displays strong acetic acid resistance and the ability to produce acetic acid and cellulose. For this reason, it is used industrially in the production of balsamic vinegar and fruit vinegar but is considered a harmful microorganism in wine as it tends to form films on the surface of grape wine, leading to oxidative off-flavors [59,60]. In this study, *Komagataeibacter* and another acetic acid bacterium, *Gluconobacter*, appeared in the early fermentation stages of YQY and disappeared when fermentation proceeded. The smell of acetic acid was not detected in YQY samples according to sensory analysis, thus their negative effects on the wine quality may be limited.

Given that the linear relationships (Spearman’s rank correlation) between the three bacterial genera (*Pantoea*, *Rhodococcus*, and *Komagataeibacter*) and VOCs/ethanol content did not show a consistent pattern among sub-regions, we performed a nonlinear fitting based on the Logistic equation (with Equation (4)) and Hill’s equation (Equation (5)).
(4)$$y=\frac{A_{1}-A_{2}}{1+{\left(x/x_{0}\right)}^{p}}+A_{2}$$
(5)$$y=V_{\max}\times \frac{x^{n}}{\left(k^{n}+x^{n}\right)}$$

The logistic fitting (Appendix A) showed that *Pantoea* was nonlinearly positively correlated with isoamyl esters, straight-chain fatty alcohols, and sulfides, while *Rhodococcus* was nonlinearly positively correlated with isobutyl and isoamyl esters at all sub-regions, but did not reach the significance level. Hill’s equation (Appendix A) demonstrated that *Pantoea* had nonlinear positive correlations with straight-chain fatty alcohols, aromatic aldehydes, and fatty acids, and *Rhodococcus* had nonlinear positive correlations with fatty aldehydes and aromatic aldehydes, again, not all reached significance levels. For *Komagataeibacter*, there was no consistent pattern among sub-regions for both equation fittings. In addition, the non-linear fitting results between *Pantoea*, *Rhodococcus*, or *Komagataeibacter* and ethanol did not show a consistent pattern among different sub-regions (Appendix A).

## 5. Conclusions

In this study, *Pantoea*, *Lactobacillus*, *Rhodococcus*, *Fructobacillus*, and *Komagataeibacter* were provisionally identified as the five core bacterial genera in the spontaneous fermentation process of ‘Cabernet Sauvignon’ grape wines from the Eastern Foot of Helan Mountain in Ningxia. However, grapes from other sub-regions and different vintages ought to be obtained for further investigation in Ningxia. *Lactobacillus* was related to the synthesis of methyl and isobutyl esters, which were beneficial as they enhance the fruity characteristics of grape wine. *Fructobacillus* contributed to the production of aromatic alcohols, improving the floral flavor of the wine. In addition, *Lactobacillus* and *Fructobacillus* were tolerant to ethanol and may be applied in mixed-culture fermentation with yeast in the future. The culture-dependent methods will be applied to verify the functions of the two genera.

## Figures and Tables

**Figure 1 foods-11-02775-f001:**
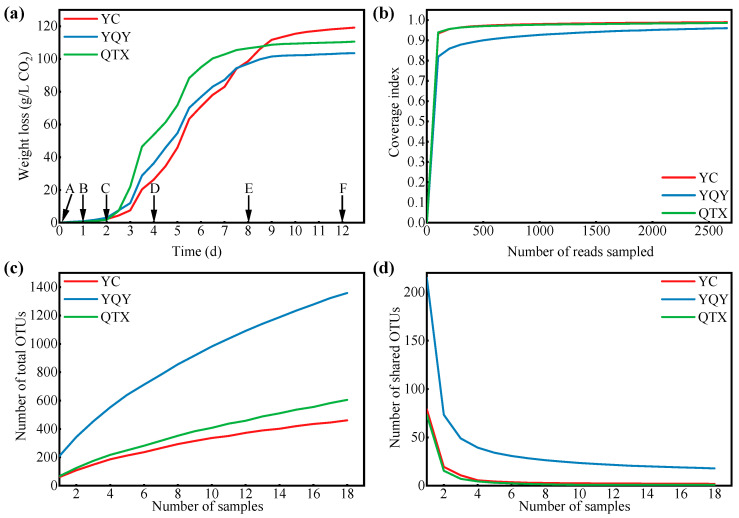
The fermentation process and sequencing evaluation. (**a**) The release of CO_2_ of three spontaneous fermentations. Samples were collected at six points marked with A (0 d), B (1 d), C (2 d), D (4 d), E (8 d), and F (12 d); (**b**) the rarefaction curve: *X*-axis, sequenced reads per sub-region; *Y*-axis, the percentage of discovered bacteria to predicted total bacteria per sub-region; (**c**) the observed numbers of bacteria per sub-region; (**d**) the shared numbers of bacteria among three sub-regions. OTUs, operational taxonomic units. Sequences with more than 97% similarity were classified as one OTU.

**Figure 2 foods-11-02775-f002:**
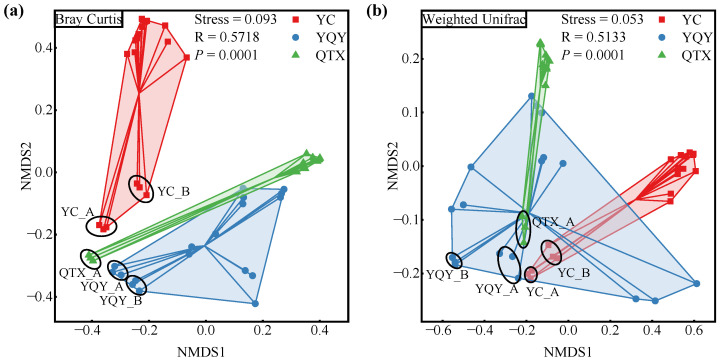
NMDS plots of samples from different sub-regions. Discrimination by (**a**) taxonomic distance, and (**b**) phylogenetic distance. The NMDS stress value below 0.1 and simultaneously *p*-value below 0.05 indicate significant discrimination among different sub-regions. The color of each square/dot/triangle corresponds to the color of each sub-region in the legend.

**Figure 3 foods-11-02775-f003:**
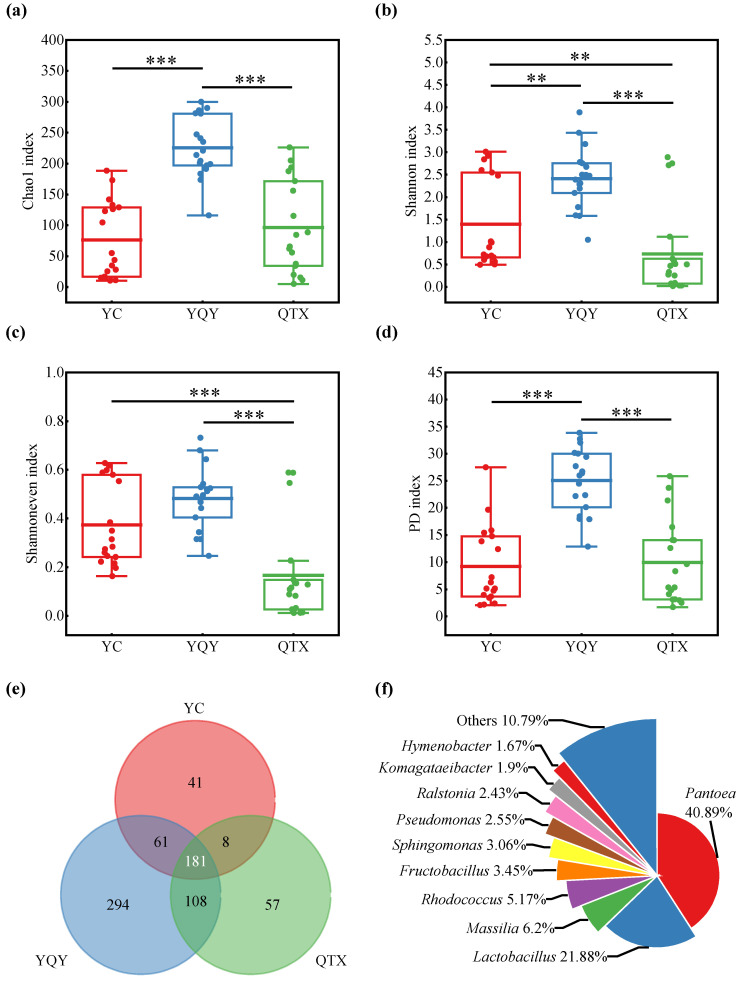
Bacterial diversity of different sub-regions. Alpha diversity at genus level: (**a**) Chao 1 diversity, (**b**) Shannon diversity, (**c**) Shannoneven diversity, and (**d**) Faith’s phylogenetic diversity; Alpha diversity values were calculated based on rarefied data established using 16S sequencing reads from 18 samples per sub-region; the statistical significance analysis was based on Wilcoxon rank-sum test, ** *p* < 0.01, *** *p* < 0.001; (**e**) the Venn plot of bacteria at the genus level; (**f**) the shared top 10 bacteria with a relative abundance of more than 1% at the genus level among three sub-regions.

**Figure 4 foods-11-02775-f004:**
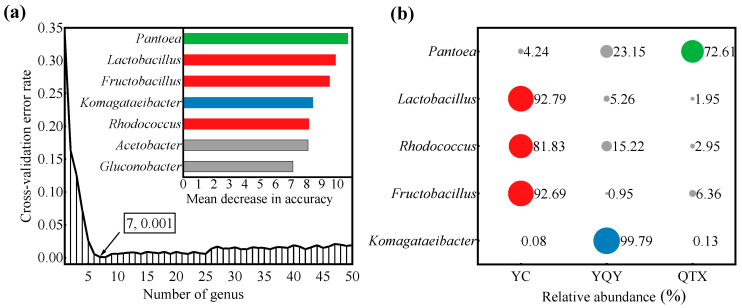
Core bacteria at the genus level of the Eastern Foot of Helan Mountain. (**a**) A RF model based on a 10-fold cross-validation test was constructed with 500 trees, using taxonomic assignments of genera as predictors and regional origin as class labels. When the bacterial number at the genus level was 7, the RF error reached the lowest value (0.001). The bar plot of variable importance at genus level: *X*-axis, bacterial importance measurement/standard deviation; *Y*-axis, bacterial name; (**b**) the percentages of the five core bacteria in three sub-regions. The color of each dot corresponds to the color of each bar in Figure 4a. The size of the circle represents the proportion of each bacterial genus in different sub-regions.

**Figure 5 foods-11-02775-f005:**
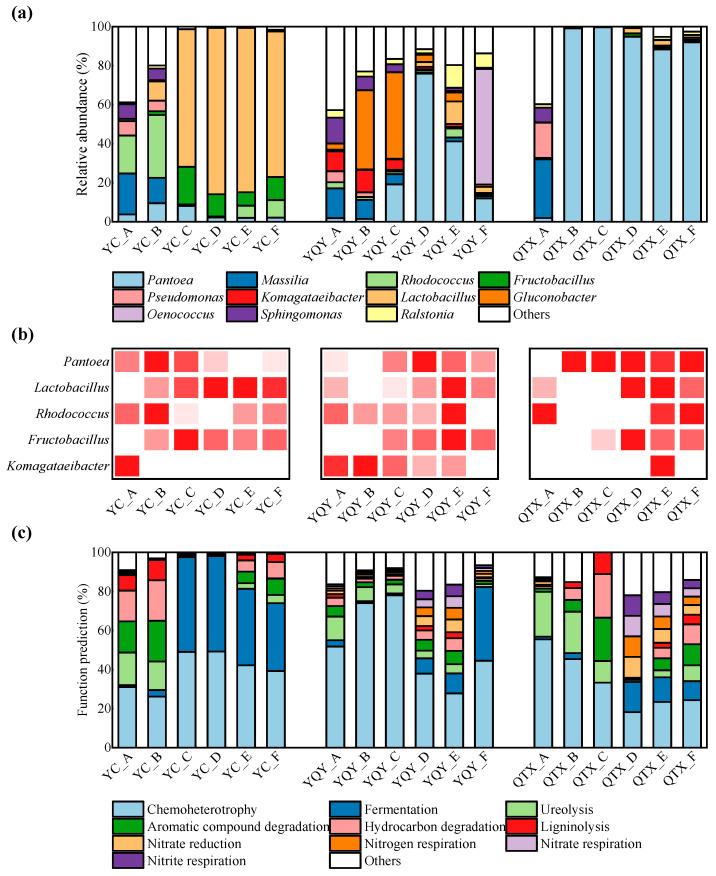
Bacterial communities and their functional annotation in different fermentation stages. The successions of bacteria at genus level during fermentation: (**a**) top 11 genera with relative abundance greater than 0.1%, and (**b**) 5 core genera (darker red squares represent higher relative abundance); (**c**) top 10 functions annotated by the FAPROTAX database.

**Figure 6 foods-11-02775-f006:**
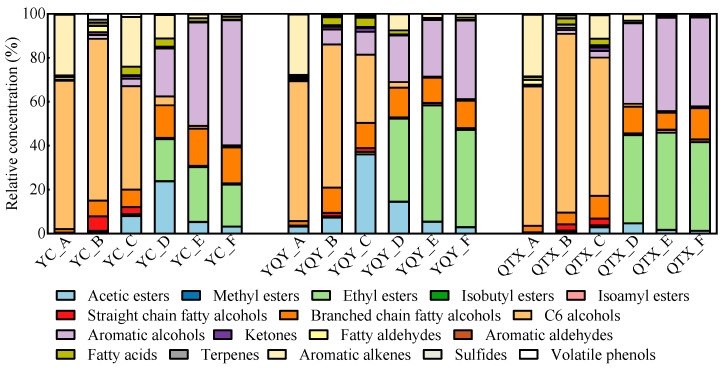
Relative concentrations of VOCs in different fermentation stages of different sub-regions.

**Figure 7 foods-11-02775-f007:**
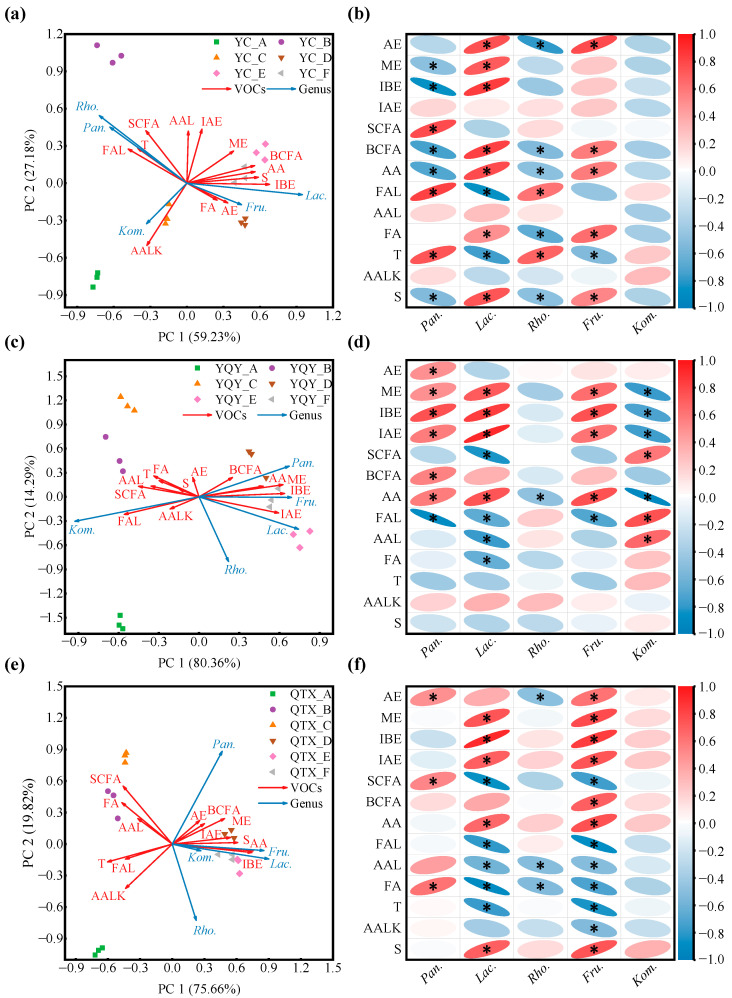
Correlation analysis between core bacteria and VOCs. RDA was used to summarize linear relationships between core bacteria and VOCs in (**a**) YC sub-region, (**c**) YQY sub-region, and (**e**) QTX sub-region. Arrow angle indicates the strength of the association between variables: Acute angle indicates strong concordance; obtuse angle, weak concordance. The different colors of arrows represent VOCs and genera. The color of each point shape corresponds to the color of each fermentation stage in the legend. A correlation heatmap was used to reveal significant linear correlations between each VOC and each genus in (**b**) YC sub-region, (**d**) YQY sub-region, and (**f**) QTX sub-region. A red ellipse with an asterisk represents a significantly positive correlation (Spearman’s rank test, *p* < 0.05), while a blue ellipse with an asterisk represents a significantly negative correlation (Spearman’s rank test, *p* < 0.05). AE: acetic esters; ME: methyl esters; IBE: isobutyl esters; IAE: isoamyl esters; SCFA: straight-chain fatty alcohols; BCFA: branched-chain fatty alcohols; AA: aromatic alcohols; FAL: fatty aldehydes; AAL: aromatic aldehydes; FA: fatty acids; T: terpenes; AALK: aromatic alkenes; S: sulfides. *Pan.*, *Pantoea*; *Lac.*, *Lactobacillus*; *Rho.*, *Rhodococcus*; *Fru.*, *Fructobacillus*; *Kom.*, *Komagataeibacter*.

**Figure 8 foods-11-02775-f008:**
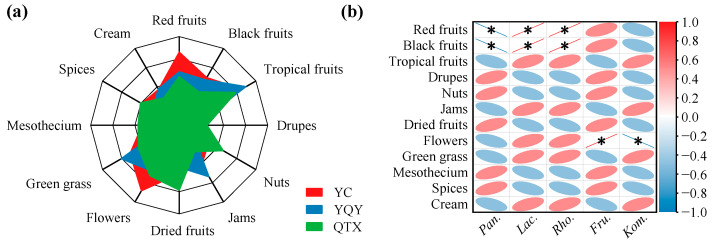
Correlation analysis between core bacteria and aroma profiles. (**a**) Aroma profile of spontaneously fermented grape wines from three sub-regions analyzed by QDA. (**b**) A correlation heatmap was used to identify significant linear correlations between each aroma descriptor and each core bacterial genus. A red ellipse with an asterisk represents a significantly positive correlation (Spearman’s rank test, *p* < 0.05), while a blue ellipse with an asterisk represents a significantly negative correlation (Spearman’s rank test, *p* < 0.05). *Pan.*, *Pantoea*; *Lac.*, *Lactobacillus*; *Rho.*, *Rhodococcus*; *Fru.*, *Fructobacillus*; *Kom.*, *Komagataeibacter*.

**Figure 9 foods-11-02775-f009:**
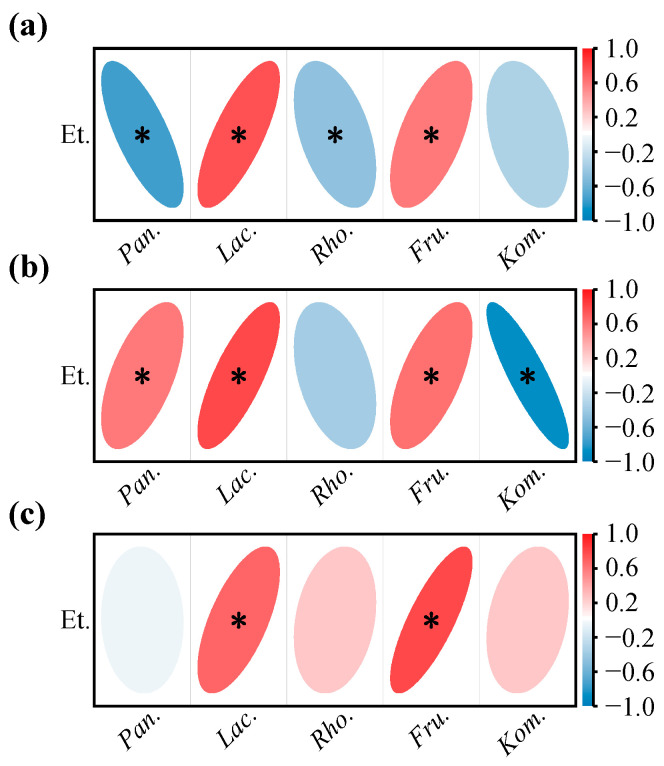
Correlation analysis between core bacteria and ethanol in (**a**) YC sub-region, (**b**) YQY sub-region, and (**c**) QTX sub-region. A red ellipse with an asterisk represents a significantly positive correlation (Spearman’s rank test, *p* < 0.05), while a blue ellipse with an asterisk represents a significantly negative correlation (Spearman’s rank test, *p* < 0.05). Et., ethanol; *Pan.*, *Pantoea*; *Lac.*, *Lactobacillus*; *Rho.*, *Rhodococcus*; *Fru.*, *Fructobacillus*; *Kom.*, *Komagataeibacter*.

**Table 1 foods-11-02775-t001:** Bacterial diversity in different fermentation stages.

Sub-Regions	Indices	Fermentation Stages						Variation
		A (0 d)	B (1 d)	C (2 d)	D (4 d)	E (8 d)	F (12 d)	
YC	Chao 1	240.36 ± 53.42	178.97 ± 34.34	22.22 ± 11.42	15.50 ± 4.92	63.27 ± 40.92	178.16 ± 149.53	*
	Shannon	3.48 ± 0.06	2.93 ± 0.07	0.87 ± 0.14	0.54 ± 0.06	0.67 ± 0.07	0.92 ± 0.17	**
	Shannoneven	0.68 ± 0.02	0.61 ± 0.02	0.31 ± 0.06	0.23 ± 0.03	0.22 ± 0.02	0.25 ± 0.08	*
	PD	23.09 ± 7.52	14.92 ± 1.56	3.70 ± 0.26	2.26 ± 0.15	5.17 ± 0.21	10.02 ± 5.48	**
YQY	Chao 1	355.23 ± 59.47	321.27 ± 49.35	312.66 ± 35.94	292.81 ± 148.86	423.41 ± 82.85	468.38 ± 34.11	
	Shannon	3.93 ± 0.39	2.81 ± 0.14	2.43 ± 0.11	1.55 ± 0.41	2.87 ± 0.16	2.11 ± 0.39	**
	Shannoneven	0.72 ± 0.04	0.54 ± 0.01	0.48 ± 0.01	0.30 ± 0.05	0.52 ± 0.02	0.39 ± 0.07	**
	PD	30.47 ± 8.38	23.73 ± 5.71	21.52 ± 2.46	26.04 ± 11.29	35.80 ± 7.12	33.62 ± 2.91	
QTX	Chao 1	291.35 ± 53.49	34.48 ± 17.27	14.28 ± 2.56	92.28 ± 34.46	221.93 ± 96.94	209.38 ± 67.09	*
	Shannon	3.46 ± 0.06	0.07 ± 0.02	0.02 ± 0.01	0.30 ± 0.04	0.81 ± 0.36	0.53 ± 0.06	**
	Shannoneven	0.66 ± 0.02	0.03 ± 0.01	0.01 ± 0.00	0.09 ± 0.02	0.17 ± 0.05	0.13 ± 0.01	**
	PD	22.65 ± 4.23	3.60 ± 0.97	2.37 ± 0.62	5.03 ± 0.82	18.91 ± 12.79	12.59 ± 2.39	**

Note: the statistical significance analysis was based on the Kruskal–Wallis H test: * *p* < 0.05, ** *p* < 0.01.

**Table 2 foods-11-02775-t002:** VIF analyses of VOCs.

VOCs	VIF Value	
	Before Filtration	After Filtration
Acetic esters (AE)	2091.5	3.1
Methyl esters (ME)	11.7	7.3
Ethyl esters (EE)	9311.0	>10
Isobutyl esters (IBE)	12.5	3.5
Isoamyl esters (IAE)	16.7	5.8
Straight-chain fatty alcohols (SCFA)	137.4	5.3
Branched-chain fatty alcohols (BCFA)	561.8	4.6
C6 alcohols (C6A)	24,228.7	>10
Aromatic alcohols (AA)	8441.4	8.4
Ketones (K)	24.8	>10
Fatty aldehydes (FAL)	43.9	8.0
Aromatic aldehydes (AAL)	8.5	6.9
Fatty acids (FA)	50.8	6.6
Terpenes (T)	7.5	6.2
Aromatic alkenes (AALK)	2579.8	6.9
Sulfides (S)	7.1	6.4
Volatile phenols (VP)	NA	>10

Note: NA indicates that the VIF value cannot be calculated by R packages.

**Table 3 foods-11-02775-t003:** Mantel test between core bacteria and VOCs.

	Euclidean Distance		Manhattan Distance	
	Mantel Coefficient	*p*-Value	Mantel Coefficient	*p*-Value
Before filtration	0.148	0.002	0.120	0.003
After filtration	0.166	<0.001	0.137	<0.001

## Data Availability

Data are contained within the article or Appendix A.

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
