# Peer review of "Bacterial Communities Related to Aroma Formation during Spontaneous Fermentation of ‘Cabernet Sauvignon’ Wine in Ningxia, China"

_foods, 2022, doi:10.3390/foods11182775_

Round 1
Reviewer 1 Report
The manuscript “Bacterial communities related to aroma formation during spontaneous fermentation of ‘Cabernet Sauvignon’ wine in Ningxia, China” presents an interesting proposal by proposing to make a correlation between the bacteria present in wine and the chemical compounds responsible for the flavor of the drink. After the evaluation of the work, the following suggestions for the authors are:
- In the keywords I suggest replacing aroma with flavor.
- In the introduction (lines 35-39) different contributions of yeasts and filamentous fungi in wine fermentation are mentioned. Discuss which are the main genera found, in addition to mentioning the main volatile organic compounds produced by yeasts.
- Line 45: “However, there are many other kinds of bacteria in the wine fermentation system whose evolutionary patterns during AF and effect on wine aroma are not clear” discuss what bacterial groups these are.
- Line 223: Is the great exclusivity of genera found (294) related to the “microbial terroir” mentioned in the manuscript? To discuss.
-Line 227: The genus Pantoea showed a high relative abundance in the study. Is this genre characteristic of some wine regions? Does the predominant presence of this group depend on the isolation method used in the studies? Compare with other studies, mentioning the identification methods used.
- Lines 514-515: Pantoea showed a positive correlation with important wine volatile compounds. Are there studies in the literature that discuss this participation?
- Conclusion: I believe that the authors could better explore the results found. Explain the use of the term “provisionally identified” for the genres identified in the study. In the abstract it was mentioned that Lactobacillus and Fructobacillus were ethanol tolerant, suggesting a future study of fermentation using mixed cultures. This result and perspective could be addressed in the conclusion.
Reviewer 2 Report
Lines 34-46. Please add some discussion here regarding the importance of indigenous microorganism in wine (microbial terroir). Use relevant reviews like Lappa, I. K., Kachrimanidou, V., Pateraki, C., Koulougliotis, D., Eriotou, E., & Kopsahelis, N. (2020). Indigenous yeasts: emerging trends and challenges in winemaking. Current Opinion in Food Science, 32, 133-143. https://doi.org/10.1016/j.cofs.2020.04.004
Line 103. It is better to use days (in my opinion)
Figure 1. Please clarify OTUs. It is expected each table/figure to be read without reading the text of the manuscript.
It is expected that the use of references in the results section to be limited. Also in my opinion the results section should be reduced since a separate section of discussion follows.
Line 448. “logarithmic period” for whom? Yeast or bacteria?
Table S2. According to the results the rate of MLF was different in each wine. Why? Please add some discussion in the text. For example in QTX almost no MLF was conducted compared to the others. It would be better to have the same data in other time intervals and not only at the end of fermentation.
The manuscript is very interesting, however, some improvements are necessary in the presentation of the results. In my opinion the section of results should be reduced and more discussion with relevant references is needed in the section of discussion.
